# Management of Gastro-Intestinal Toxicity of the Pi3 Kinase Inhibitor: Optimizing Future Dosing Strategies

**DOI:** 10.3390/cancers15082279

**Published:** 2023-04-13

**Authors:** Claire Breal, Frederic Beuvon, Thibault de Witasse-Thezy, Solene Dermine, Patricia Franchi-Rezgui, Benedicte Deau-Fisher, Lise Willems, Eric Grignano, Adrien Contejean, Didier Bouscary, Jean Luc Faillie, Jean-Marc Treluyer, Corinne Guerin, Laurent Chouchana, Marguerite Vignon

**Affiliations:** 1Service d’Hématologie Clinique, Hôpital Cochin, Assistance Publique-Hôpitaux de Paris, Centre-Université de Paris, 75014 Paris, France; 2Anatomopathologie, Hôpital Cochin, Assistance Publique-Hôpitaux de Paris, Centre-Université de Paris, 75014 Paris, France; 3Centre Régional de Pharmacovigilance, Service de Pharmacologie, Hôpital Cochin, AP-HP, Centre-Université de Paris, 75014 Paris, France; 4Gastro-Entérologie, Hôpital Cochin, Assistance Publique-Hôpitaux de Paris, Centre-Université de Paris, 75014 Paris, France; 5Centre Régional de Pharmacovigilance, CHU Montpellier, IDESP, INSERM, University of Montpellier, 34193 Montpellier, France; 6Pharmacie, Hôpital Cochin, Assistance Publique-Hôpitaux de Paris, Centre-Université de Paris, 75014 Paris, France

**Keywords:** idelalisib, phosphatidylinositol 3-kinase delta (PI3Kδ) inhibitor, colitis, pharmacovigilance, adverse drug reaction

## Abstract

**Simple Summary:**

Targeted therapies are becoming more widespread in the treatment of indolent B-cell malignancies, as has been the case for chronic lymphocytic leukemias and small B-cell non-Hodgkin lymphomas. Due to the severity of these diseases, several drugs have received fast track approval since 2014 including the BTK inhibitor ibrutinib, the Bcl2 inhibitor venetoclax and various P13 kinase inhibitors. Of the Pi3 kinase inhibitors, idelalisib was the first of this class to be approved, followed by the second-generation drugs copanlisib, duvelisib and umbralisib. However, the last of these agents have now been withdrawn, partly due to severe gastrointestinal side effects. We herein describe these gastrointestinal effects, as reported in clinical trials, and then review real-world data for these targeted inhibitors using world-wide pharmacovigilance evidence. Our own single-center experience of 15 patients with an indolent B-cell malignancy, from with six biopsies were derived, has helped us to describe the incidence and severity of Pi3 kinase-induced colitis. Histological analysis of these cases yielded clues toward an immunopathological hypothesis. We finally speculate on future directions in relation to managing this severe toxicity and thereby optimizing the safe use of these drugs.

**Abstract:**

The phosphatidylinositol 3-kinase (PI3K) pathway plays a key role in cancer progression and in host immunity. Idelalisib was the first of this class to be approved with the second-generation Pi3 kinase inhibitors copanlisib, duvelisib and umbralisib, subsequently being approved in the United States. Real-world data are lacking, however, in relation to the incidence and toxicity of Pi3 kinase inhibitor-induced colitis. We here review, in the first instance, the general landscape of the Pi3K inhibitors in the context of hematological malignancies, with a focus on the adverse gastrointestinal side effects reported by various clinical trials. We further review the available worldwide pharmacovigilance data in relation to these drugs. Finally, we describe our own real-world experience with idelalisib-induced colitis management in our center and in a national setting.

## 1. Introduction

The phosphatidylinositol 3-kinase (PI3K) pathway plays a key role in both cancer and immunity [1]. Among the three classes of PI3Ks, class I is the most important for the development, differentiation and activation of B and T cells. This class has thus become a target for developing treatments of B-cell lymphoproliferative disorders. Four PI3K class I isoforms are distinguishable: PI3Kα, PI3Kβ, PI3Kδ and PI3Kγ. The delta and gamma isoforms are preferentially expressed on leucocytes [1] and every approved Pi3 kinase inhibitor inhibits the delta isoform. Four Pi3k inhibitors have been approved by the US FDA since 2014: The first of these was the PI3Kδ inhibitor idelalisib (zydelig/GS-1101/Gilead) for the treatment of relapsed B-cell non-Hodgkin lymphoma (B-NHL) [2] and chronic lymphocytic leukemia (CLL) [3], approved both in the US and in Europe. This was followed by a fast track approval in 2017 in the US of the pan-class PI3K inhibitor copanlisib (Aliqopa/BAY 80-6946; Bayer) [4] and in 2018 of the dual PI3Kδ/PI3Kγ inhibitor duvelisib (Copiktra/IPI-145/INK1197; Verastem, now Secura Bio) [5] for the same indications. Lastly, the PI3Kδ and casein kinase-1ε (CK1ε) inhibitor umbralisib (Ukonic/TGR-1202; TG therapeutic) was approved in 2021 in ISfor the treatment of follicular lymphoma (FL) and marginal zone lymphoma (LZM) after two prior lines of therapy had been given [6]. 

Notwithstanding their clinical approval, safety warnings have curtailed the wider use of these promising agents. Delayed immune-mediated toxicities from the use of Pi3K inhibitors, including colitis, hepatitis, rash and interstitial pneumonitis, are the principal adverse effects that raise concerns for clinicians. Diarrhea has been found in different studies to be the most common complication of these drugs that has led to dose reductions or treatment discontinuations. The aim of our present study was to report on the risk of Pi3K inhibitor-induced colitis and clarify its clinical presentation. We contend that efforts should be made to optimize the use of this drug class. 

In the first instance, we provide an overview of Pi3K inhibitor development in the context of hematological malignancy. Despite the initial accelerated approval of these agents in 2014, serious adverse events and lack of any confirmatory trials lead to several voluntary withdraw indications from the FDA in two years. We focus herein on the frequency of gastrointestinal side effects reported in clinical trials. Second, we report on the currently available worldwide pharmacovigilance data on the gastro-intestinal toxicity arising from the use of Pi3K inhibitors. We address how safety concerns continue to limit the use of the new classes of these drugs. Finally, we describe our own real-world experiences with idelalisib-induced colitis management at our center as well as in a national setting in France. This experience has provided some important insights into the management of this severe toxicity and possible optimization of future dosing regimens and clinical trials.

## 2. Material and Methods

### 2.1. Literature Review

We conducted a literature review of the development and approval history of Pi3K inhibitors in the US and in Europe since 2014 as well as of the main clinical trials of these agents to date. 

### 2.2. Review of the WHO Global Safety Database

We reviewed VigiBase (https://www.who-umc.org/vigibase/vigibase/, accessed on: 19 January 2022), the World Health Organization (WHO) global safety database, as part of this current study. This database is maintained, deduplicated and deidentified by Uppsala Monitoring Centre (Uppsala, Sweden). VigiBase is the world’s largest continuously updated pharmacovigilance database and contains over 23 million reports of suspected adverse drug reactions gathered from national pharmacovigilance systems. Researchers and clinicians from more than 130 countries, over five continents, submit spontaneous safety reports to this repository. Information concerning the reporter, patient, suspected and concomitant drugs, and suspected adverse drug reactions and their seriousness, is provided for each case. We conducted disproportionality analysis to assess the associations between colitis reporting and PI3K inhibitors used (idelalisib, alpelisib, duvelisib, copanlisib or umbralisib). The pharmacovigilance statistical approach we used was based on a case/non-case study that is similar to case–control studies, but designed for the purpose of pharmacovigilance estimates. This approach has a proven efficacy in identifying new adverse drug reactions [7]. It estimates whether an adverse event (i.e., colitis) is differentially reported for a drug (i.e., either a specific PI3K inhibitor or the whole drug class) compared to other agents, using the reporting odds ratio (ROR) for each drug-adverse event combination. Cases of this nature include reactions reported under the high-level term “non-infectious colitis” according to MedDRA, whereas non-cases are reported as any other adverse drug reaction (ADR).

### 2.3. Single-Center Experience

Patients receiving idelalisib between 2014 and 2019 at our hospital were identified using our institutional pharmacy records (Cochin Hospital, Hematology Department, Paris, France). The standard idelalisib regimen used at our center was a 150mg dose twice daily until disease progression occurred or the patient developed a severe adverse reaction to the drug. The medical charts of our patients receiving idelalisib were retrospectively reviewed to identify colitis cases. Idelalisib-induced colitis was defined in accordance with an expert statement [8] as onset of diarrhea (≥3 soft or liquid stools by day) at least 8 weeks after the commencement of idelalisib treatment, and in the absence of another infectious etiology that could explain this symptom (e.g., negative results from a standard stool examination for *Clostridium difficile* or a positive cytomegalovirus blood RNA assessment). The severity of diarrhea was graded using common terminology criteria for adverse events (CTCAE) [9]. Digestive biopsies were retrospectively evaluated by a senior pathologist (FB) and were systematically stained for CMV to exclude opportunistic infections. 

### 2.4. Review of the French Pharmacovigilance Database

We performed a retrospective nationwide analysis of idelalisib-induced colitis cases that had been reported to the French Pharmacovigilance Network during the same period as our current study from 2014 to 2019. This national network of 31 regional pharmacovigilance centers, established among French university hospitals in 1986, receives and analyses spontaneous reports of suspected ADRs. Each individual case safety report (ICSR) is classified as “non-serious” or “serious”, based on WHO criteria such as death, life-threatening conditions, requiring/prolonging hospitalization or persistent or significant disability/incapacity outcomes [10]. Causality assessments in this database are made by clinical pharmacologists based on a national scoring system [11]. Adverse drug reactions are reported in accordance with the hierarchical Medical Dictionary for Regulatory Activities (MedDRA) (https://www.meddra.org/, accessed on 19 January 2022). We screened for and collected ICSRs that were indicative of idelalisib treatments and resulting gastrointestinal disorders (according to the System Organ Class). A case-by-case review was then conducted by a junior and senior clinical pharmacologist (TdW, LC). Idelalisib-induced colitis was identified in accordance with the same criteria detailed above. As second-generation Pi3K inhibitors are not currently approved for use in France, we did not extend our search to other drugs.

### 2.5. Ethics Approval

This study was approved by our local Institutional Review Board (IRB registration #00011928) and was designed in accordance with the Good Clinical Practice Guidelines and the Declaration of Helsinki. ICSRs in the French pharmacovigilance database are fully anonymized and made available to pharmacovigilance centers.

### 2.6. Statistics

The current study data are presented as a number (%) or as a median value (interquartile rage, IQR). The ROR and 95% confidence interval (CI) were calculated in each case as follows
ROR [95% CI] = *ad*/*bc* [*e* ± 1.96√(1*a* + 1*b* + 1*c* + 1*d*]
where *a* is the number of colitis cases reported with a specific PI3Ki, *b* is the number of non-cases (i.e., all other adverse drug reaction reports) reported with a specific PI3Ki, *c* is the number of colitis cases reported with any other drugs and *d* is the number of non-cases (i.e., all other adverse drug reaction reports) reported with any other drugs in the database [12]. A lower or higher boundary 95% confidence interval <1 or >1, respectively, is deemed significant as for OR interpretation. Statistical analysis was performed using Microsoft Excel^®^.

## 3. Results

### 3.1. History of Pi3 Kinase Inhibitor Development

The US FDA and European Medicines Agency (EMA) granted approval for the first Pi3K inhibitor, idelalisib, in 2014. This agent received approval in association with the anti-CD20 drug rituximab for relapsed CLL and gained an accelerated approval for the treatment of FL and small lymphocytic leukemia (SLL). Notably, however, the FDA raised a safety alert for idelalisib in 2018 due to infection and immune-mediated toxicity associated with its use. In February 2022, the FDA withdrew FL and SLL indications for this drug due to a lack of evidence of its clinical benefits for these diseases. A randomized clinical trial of idelalisib in the treatment of FL and SLL reported increased toxicity and death [13] caused by adverse events and there has been no evidence of a better progression-free survival rate, translating into an improved overall survival outcome, with the use of this drug for these low-grade indolent disorders [14]. However, PI3K inhibitors remain the best option after BTK inhibitors or venetoclax in the management of CLL, as per ESMO guidelines [15].

A second Pi3K inhibitor, copanlisib, was approved in 2017 and is administered intravenously on a weekly schedule. A single-arm phase I trial of this agent had reported a 59% response rate in patients with relapsed or refractory indolent lymphoma [16] and copanlisib was then granted fast track approval by the US FDA. However, in the subsequent randomized trial CHRONOS-3 [4], the association of copanlisib and rituximab versus placebo and rituximab failed to show a better overall survival, despite a better progression-free survival. The indications for the use of copanlisib were withdrawn voluntarily from the FDA in December 2021.

A third Pi3K inhibitor, duvelisib, received FDA approval in 2018 for CLL patients who have received at least two prior therapies. Approval by the EMA followed in 2021 [5]. Based on a phase II trial, fast track approval was also granted for this drug for refractory indolent non-Hodgkin lymphoma in the US [17], but it was withdrawn in December 2021 due to the absence of a confirmatory trial. 

Lastly, the PI3K inhibitor umbralisib received approval in February 2021 for follicular and marginal-zone lymphoma [6]. However, an increasing imbalance in overall survival was found with this therapeutic, despite a higher progression-free survival, that suggested death related to adverse events [18]. The FDA then placed a partial clinical hold on trials using umbralisib and anti-CD20 ublituximab as treatments for CLL and *B-NHL*. This led to a voluntary withdrawal of the approbation. 

This complex history of Pi3K inhibitor development highlights the encouraging efficacy of these agents but also their severe toxicity profile [19]. It is therefore of interest to improve our knowledge of the specific side effects of these drugs [20]. Induced colitis is particularly clinically important in this setting, as phase II and phase III results for Pi3K inhibitors have included an incidence of grade 3 diarrhea of 15%, which is one of the main reasons for treatment discontinuation.

### 3.2. New Pi3K Inhibitors and the Risk of Colitis

#### 3.2.1. Literature Review

A randomized, double-blind, phase III study of idelalisib plus rituximab versus placebo plus rituximab in patients with relapsed chronic lymphocytic leukemia (CLL) has indicated a superior efficacy of the idelalisib arm such that patients could enroll in an extension study to receive idelalisib monotherapy. The long-term efficacy and safety of idelalisib was assessed in a prior study of 110 patients who received at least one dose and grade 3 or 4/5 was observed at frequencies of 10.9% and 8.2%, respectively, with increased incidence over time [21]. Colitis seemed to affect younger patients in this trial with less prior exposure to treatment. Two-year follow-up analyses of the phase-1 CHRONOS study of copanlisib and rituximab in the context of indolent lymphoma reported a median safety follow-up of 6.7 months (26% of patients received treatment for > 1 year). The grade 3 diarrhea incidence was 8.5% [22]. The safety of duvelisib was reported through the DUO trial comparing this agent with ofatunumab. The most frequent grade 3/4 treatment-emergent adverse events were diarrhea (23%), often leading to treatment discontinuation [23]. In a phase IIb open-label study evaluating umbralisib in indolent lymphoma cases after one or two lines of treatment, monotherapies led to grade ≥ 3 diarrhea in 10.1% of patients [6]. Other integrative analysis of four different trials reported a grade 3 diarrhea rate of 7.3% and non-infectious colitis of 2% [24].

It must be noted that these aforementioned data are available from the current literature on a very select population of patients that were eligible for a clinical trial. Real-world clinical data provided by pharmacovigilance are therefore necessary. 

#### 3.2.2. Pharmacovigilance Database

Clinical trials are conducted in bespoke populations and have short follow-up durations. Real-world clinical data are therefore necessary to properly evaluate the toxicity of the new drugs being tested. Of the 29,334,896 spontaneous reports in VigiBase up to 19 January 2022, 11,143 included a PI3K inhibitor as the suspected drug in relation to the adverse event. Among these, 342 colitis cases could be identified, including 296 with idelalisib, 27 with alpelisib, 16 with duvelisib, 3 with copanlisib and 1 with umbralisib. The reports mainly originated from healthcare providers (*n* = 304, 89%), in the US (*n* = 100, 29%), the UK (*n* = 41, 12%) and France (*n* = 39, 11%). They concerned male patients in 163 (48%) cases. The median age (IQR) of the affected patients was 69 (range, 59–75) years and these cases were mostly serious (*n* = 297, 87%) with 34 (10%) involving a fatal reaction outcome. Compared to other drugs, we found a large increase in colitis reporting with PI3K inhibitors (ROR, 95% CI: 9.5 [8.6–10.6]) (Table 1). This increase was particularly more marked for idelalisib and duvelisib (PI3Kδi). Regarding umbralisib, no conclusion could be drawn due to insufficient data.

### 3.3. French Experience of Idelalisib-Induced Colitis

#### 3.3.1. Single-Center Experience

Fifteen patients (nine men and six women) were treated with idelalisib at our hospital between 2014 and 2019, comprising eight B-NHL cases and seven CLL patients. The clinical histories of these patients are provided in Table 2.

WM, Waldenström macroglobulinemia; CLL, chronic lymphocytic leukemia; Tts, treatments.Six (40%) of our patients developed idelalisib-induced colitis with a median time to onset of 3.5 (IQR, 3–5.5) months (Table 2). These cases had no history of digestive disorders or concomitant medications (non-steroidal anti-inflammatory drugs, angiotensin receptor blockers or proton pump inhibitors) known to have gastrointestinal side effects. No drug interactions with idelalisib were identified. All six cases developed choleriform diarrhea without abdominal pain, fever or bleeding and were severe (grade ≥ 3). 

Idelalisib was withdrawn in all six cases at our hospital that developed colitis. In two patients, the colitis improved with symptomatic treatment and idelalisib discontinuation. Glucocorticoid therapy was introduced in four patients. After resolution of the diarrhea, idelalisib therapy was continued with a dose reduction (100 mg daily or twice daily) for three patients, combined with steroid medication. However, this idelalisib rechallenge led to colitis relapse in two patients, with a fatal outcome in one case. Three other patients died of disease progression in the absence of other therapeutic options after idelalisib discontinuation. This idelalisib colitis history is outlined in Table 2. 

#### 3.3.2. Analysis of the French Pharmacovigilance Database

Over the same period as our present study, 56 cases related to gastrointestinal disorders caused by idelalisib were reported to the French Pharmacovigilance Network. After a case-by case review of this cohort, ten cases were excluded from further analysis, five due to non-colic symptoms, four due to bacteriologic findings in the stools that could explain the symptoms (three patients with *Campylobacter jejuni* and one with *Clostridium difficile)* and one case with symptoms that were not compatible with idelalisib therapy. A final total of 46 colitis cases from this database were analyzed herein (32 men and 14 women). Their median age was 71 (range, 66–78) years and their clinical characteristics are listed in Table 3. The median time to the onset of diarrhea was 122 days (range, 74–212) after commencing idelalisib treatment. There was no other medication that could have led to the onset of colitis in 38 (83%) patients. A large majority of these patients had watery diarrhea (90%) and the symptoms were observed to have been serious in 43 cases (93.5%). A colonoscopy was performed in 19 (41.3%) cases, which at a macroscopic level revealed colonic injuries for 16 (34.8%) patients and normal findings in the remaining 3 (6.5%) patients. 

In terms of clinical interventions for idelalisib-induced colitis among the French Pharmacovigilance Network cases, the median period between the first symptoms and corrective action was 20 days (range, 6–30). Idelalisib was discontinued in most cases (43 patients, 93%) and 23 patients (50%) received corticosteroids. The median time to symptom resolution was three days in patients who received steroids and 10 days in patients who did not. Three patients were maintained on idelalisib at a reduced dose and had a favorable outcome. 

#### 3.3.3. Histological Description

All six cases from our center manifesting idelalisib-induced colitis underwent a colon biopsy. The histologic patterns that were observed from the sampled tissues ranged from mild lymphocytic colitis to acute mucosal necrosis. As shown in Figure 1, these cases showed a varying extent of different abnormalities, including the following:-Increased lamina propria inflammation with lymphocytes, plasma cells, neutrophils and eosinophils;-Acute or chronic ischemic-like lesions with an atrophic or withered crypt appearance, including hyalinization of the lamina propria;-Apoptosis at the bases of the crypts with lymphocyte infiltration of the crypt epithelium, similar to what is seen in a graft vs. host disease but surrounded by more inflammatory cells.

Furthermore, two patients had crypt abscesses that were reminiscent of active ulcerative colitis. These findings are consistent with the pharmacovigilance data, i.e., histologic analysis indicated an inflammatory chorion with a predominance of lymphocytes and plasma cells in some patients, but also inflammatory colitis with the presence of apoptotic bodies. 

## 4. Discussion

This review focuses on the gastrointestinal toxicity of Pi3K inhibitors and discusses how a single-center experience combined with larger pharmacovigilance data can help to manage this severe complication. By analyzing the French and the WHO global pharmacovigilance databases, we have provided crucial new information on the risk of gastrointestinal complication of several Pi3K inhibitors such as idelalisib, copanlisib and duvelisib. These data are complementary to the evidence provided from our literature review. 

Pi3K inhibitor-induced colitis is typically late onset (median delay of 4 months) and is serious in most instances (grade ≥ 3). The affected patients display a non-specific inflammatory colitis profile with no infectious cause which causes an interruption of the idelalisib therapy in most cases. Analysis of reports from the French pharmacovigilance database produced similar findings. Indeed, the pharmacovigilance data presented here indicated that colitis is late-occurring with a median time from the start of therapy to diarrhea onset of 122 days (105 days in our institution). Idelalisib-induced colitis was always severe in these cases with grade ≥ 3 diarrhea, which is indicative of ≥7 stools per day over baseline, incontinence, hospitalization or limitations to self-care activities during daily living in accordance with the CTCA. Furthermore, half of our current patients required intensive care due to their life-threatening status. Hence, this is a non-acceptable level of toxicity and it will necessarily impair the development of the drug. 

Idelalisib-induced colitis is defined as on onset of diarrhea (≥3 soft or liquid stools by day) at least 8 weeks after commencement of this drug treatment, and in the absence of another infectious etiology that could explain these symptoms. Early onset diarrhea is usually grade 1–2 and can managed by diet and symptomatic agents. However, the management of late-onset diarrhea requires different approaches that are based on expert panel recommendations [8]. As was performed in our present cohort, initial microbial explorations are crucial to eliminate other possible infectious causes of the diarrhea and avoid unnecessary treatment interruption. Due to the severity of the colitis in the idelalisib-treated patients in our hospital, this drug treatment was always discontinued in these cases. This cessation alleviated the diarrhea and a further intervention with steroids can reduce the subsequent recovery time [25]. As described previously by Coutré et al., budesonide, a glucocorticoid with a local anti-inflammatory mode of action, is also a useful therapeutic for avoiding systemic effects. Coutré and al. have reported a 58% success rate with an idelalisib rechallenge, in association with budesonide or corticosteroids, and with a dose reduction. Moreover, safety analysis has shown that a higher dose of PI3K inhibitor results in a higher grade 3 adverse effect, including diarrhea [26]. This drug class thus presents a narrow range between efficacy and safety, and optimal dosing is not yet certain. Another approach would be to use Pi3K inhibitors over a shorter duration. The clinical benefits of these agents seem to persist for patients with CLL who have experienced treatment interruption after receiving idelalisib for ≥6 months, supporting a fixed duration of treatment with this class of therapeutic [27]. 

One of the important findings from our current real-world analyses of idelalisib-induced colitis was the 20-day median delay between symptom occurrence and specific intervention. As second-generation Pi3K inhibitors will be used in hematology and oncology settings, the lessons from our first experience with idelalisib will be useful. This highlights the need to educate patients on possible gastrointestinal complications and provide careful follow-up management. As steroid treatment is efficient, early diagnosis could prevent serious symptoms, and help to avoid drug treatment interruptions. 

It is important to note that an interruption in the idelalisib regimen will impair the hematological prognosis in CLL and B-NHL patients, especially if there are other negative prognostic indicators and no other treatment options [28]. In this regard, three patients in our current cohort died of disease progression. Immuno-chemotherapy is not a standard treatment for CLL anymore and new targeted agents are recommended in the first line setting such as BTK inhibitors and anti bcl-2 agents [29]. Idelalisib is currently recommend in the CLL relapse setting, in the absence of other options, especially in the context of p53 mutations or deletions [15]. Novel Pi3K inhibitors will also be used in the context of low-grade lymphoma after chemotherapy failure [30]. Therefore, to improve the hematological prognosis of these patients, clinicians must learn to recognize and then manage colitis.

The histological patterns of the idelalisib-induced colitis in our present patients, as in the current literature, mimic different aspects of frequent inflammatory disorders, which points to an immune pathophysiology [31] such as lymphocytic colitis, ischemic colitis, chronic intestinal inflammatory disease and acute graft versus host disease with apoptotic bodies. T-cell deregulation, rather than a direct effect of the Pi3K inhibitor drug on colonic epithelial cells, may explain this complication [32]. A delayed onset and responsiveness to steroids in these cases support this hypothesis. The immune colitis secondary to Pi3K inhibitors seem to be specific to the delta inhibitors, especially idelalisib and duvelisib. The delta isoform of PI3K is largely limited to hematopoietic cells, while its alpha and beta isoforms have ubiquitous expression. Hence, Pi3K inhibition decreases the T-regulatory cell number and function [33,34]. In a prior study using a mouse model, it was found that the inactivation of p110δ induced an impaired B- and T-cell response and, interestingly, that these mice also developed inflammatory bowel disease [35]. The inhibition of mesenteric B cells and T regulatory cells induces the infiltration of the intestinal mucosa by CD8 cytotoxic cells [36]. The results of colon biopsies, reported by several studies [37,38], has indicated a frequent increase in intraepithelial lymphocytes following idelalisib treatment, comprising mainly T cells with a predominance of the CD8+ variant. Pi3K also has a role in the innate immune response via the alteration of cytokine production [39].

## 5. Conclusions

In conclusion, the PI3K inhibitor class is very active against lymphoma and CLL but the currently limited availability of more tolerable agents has limited its wider use. Notably, however, these diseases remain incurable and new therapeutic options are vital. Relapses after BTK and BL2 inhibitor or immunotherapy regimens represent an unmet medical need [40], but the management of toxicity is a continuing challenge with the use of PI3K inhibitors [41]. Future cooperation and collaboration between physicians and clinical pharmacologists will be essential to ensure patient safety and help to conduct further clinical trials of the PI3K inhibitor class of drugs.

## Figures and Tables

**Figure 1 cancers-15-02279-f001:**
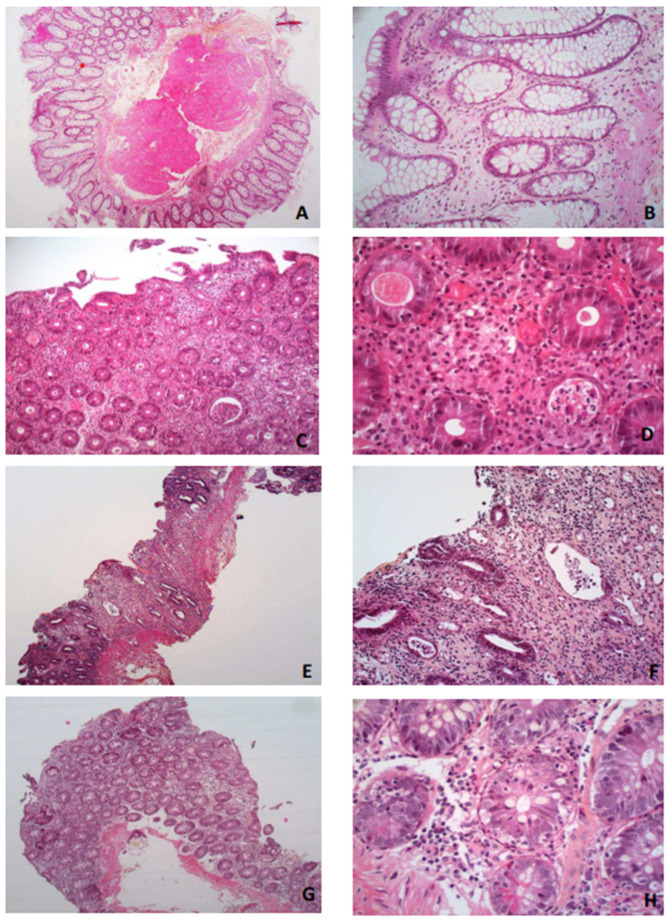
Histological patterns of the idelalisib-induced colitis in the study patients. (**A**,**B**) Mild crypt distortion without any other anomaly (case 1). (**C**,**D**) Acute inflammatory changes, crypt abscess, IBD/ulcerative colitis-like (cases 3 and 5). (**E**,**F**) Withering crypt, fibrosis of the lamina propria, chronic ischemic-like lesion (case 4). (**G**,**H**) Apoptotic predominant aspect, GVH-like (cases 2 and 6). (**A**,**E**) ×25; (**C**,**G**) ×100; (**B**,**F**) ×200; (**D**,**H**) ×400.

**Table 1 cancers-15-02279-t001:** Colitis reporting and odds ratios of reporting for PI3K inhibitors within the WHO global safety database.

Pi3Kinhibitor	Target	Colitis Cases	Non-Cases	Reporting OR (95% CI)

	-	342	10,801	9.5 (8.6–10.6)

Idelalisib	PI3Kδ	296	6093	14.6 (13.0–16.4)
Alpelisib	PI3Kα	27	4092	2.0 (1.4–2.9)
Duvelisib	PI3Kγ/PI3Kδ	16	365	13.2 (8.0–21.7)
Copanlisib	All PI3K	3	172	5.2 (1.7–16.4)
Umbralisib	PI3Kδ	1	82	-

**Table 2 cancers-15-02279-t002:** Characteristics of idelalisib-induced colitis cases from a single center in France.

	Patient Characteristics					Idelalisib-Induced Colitis Characteristics								
Case	Age (Years)	Sex	Disease	Prior Tts	Concomitant Therapy	Grade	Intensive Care Admission	Delay (Months)	Colonoscopy	Treatment Stopped	Management	Recovery	Rechallenge	Death Occurred
1	76	M	WM	6	ofatumumab	4	Yes (hypokalemia)	3	Normal colonic mucosa	Yes	Symptomatic treatment	Yes	No	Yes (disease)
2	71	M	WM	4	⁄	3	No	3	Erythema and absence of vascular pattern in the colon	Yes	Prednisone 0.8 mg/kg	Yes	Yes	/
3	76	M	CLL	6	ofatumumab	4	Yes (hypovolemic shock)	4	Erythema, absence of vascular pattern and superficial ulcerations in the rectum and colon	Yes	Prednisone 1 mg/kg	Yes	No	Yes (disease)
4	66	M	WM	1	ofatumumab	4	Yes (hypovolemic shock)	4	Erythema and absence of vascular pattern in the rectum and colon	Yes	Prednisone 0.8 mg/kg	Yes	No	Yes (disease)
5	84	F	WM	2	⁄	3	No	6	Erythema and absence of vascular pattern in the colon	Yes	Enteric budesonide	Yes	Yes (fatal colitis)	Yes (colitis)
6	89	F	CLL	2	rituximab	3	No	3	Absence of vascular pattern in the sigmoid segment	Yes	Symptomatic treatment	Yes	Yes (no colitis occurrence)	/

**Table 3 cancers-15-02279-t003:** Characteristics of the idelalisib-induced colitis cases reported to the French Pharmacovigilance Network.

Patient Characteristics	n (%)
Age—years mediane (range)	71 (65–78)
Sex—male: female	32:14
Type of hemopathy	
- CLL - B-NHL	26 (56.5%) 20 (43.5%)
Anticancer drug treatments	
- Idelalisib alone - Idelalisib associated with anti-CD20	27 (58.7%) 19 (41.3%)
Time to first digestive symptoms—days	122 (74–212)
Time to specific management—days	20 (6-30)
Colitis seriousness	
Hospitalization Life-threatening condition Death	43 (83.5%) 35 (76.1%) 4 (8.7%)

Colonoscopy findings	
- Unknown - Nonspecific inflammatory colitis - Erythematous colitis - Normal findings - Ulcerous colitis	30 9 4 3 3
Idelalisib management	
- Withdrawn - Dose reduction - Dose unchanged	43 2 1
Therapeutic management	
- Glucocorticoids - No specific drug - Symptomatic treatment - Antibiotics - Antibiotics and glucocorticoids	23 14 5 3 1

## Data Availability

The data that support the findings of this study are not publicly available due to data protection laws but are available from the corresponding author upon reasonable request.

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
