# Peer review of "Management of Gastro-Intestinal Toxicity of the Pi3 Kinase Inhibitor: Optimizing Future Dosing Strategies"

_cancers, 2023, doi:10.3390/cancers15082279_

Round 1

Reviewer 1 Report

It is an interesting review of PI3K inhibitors focusing on gastrointestinal toxicity. I think the authors did great at putting together this review. Managing the toxicities of PI3K inhibitors remains an issue, and this manuscript contributes to the field and should be published.

This submitted version of the manuscript will need to be improved before acceptance. Here are my comments: 

·      English editing and polishing: the manuscript needs to be edited thoroughly. There are some missing words, typos to be fixed, and wording to improve. Ideally, a round of professional editing shall suffice.

·      “real-life”: please do not use “real-life” but a more scientifically/medically inclined equivalent. This reviewer fully understands the meaning of it, but it should not be put like this in the manuscript.

·      Regarding PI3K gastrointestinal toxicities, what about managing early and late-onset toxicities?

·      The citations should be increased and enhanced to support their argumentation better.

Author Response

Reviewer 1 :

  • English editing and polishing: the manuscript needs to be edited thoroughly. There are some missing words, typos to be fixed, and wording to improve. Ideally, a round of professional editing shall suffice.

We agree to this remark. Please find enclosed our manuscript correction certificate

  • “real-life”: please do not use “real-life” but a more scientifically/medically inclined equivalent. This reviewer fully understands the meaning of it, but it should not be put like this in the manuscript.

Thank you for this correction. We changed « real-life » by « real-world »

  • Regarding PI3K gastrointestinal toxicities, what about managing early and late-onset toxicities?

We have discussed this important issue in our revised manuscript (l 335-337)

  • The citations should be increased and enhanced to support their argumentation better

We agree to this remark. We added citations, focusing on pathology and treatment strategy.

Reviewer 2 Report

The research article under the title “Management of gastro-intestinal toxicity of Pi3 kinase inhibitor: optimizing future dosing strategies” by Breal and coworkers presents interesting results on the use of Pi3 kinase inhibitors and their toxicity towards gastro-intestinal system. The results are presented in a clear way with statistical analysis done well. My recommendation is MINOR REVISION and the article will be suitable for publication after these questions are addressed by the authors:

1.      Simple summary – either targeted therapies or targeted therapy is

2.      Line 21 there is and without the end of sentence

3.      Line 22 – severe gastro-intestinal is not finished sentence

4.      Line 23 – datas should be changed to data

5.      Lin 24 – give should be changed to gives

6.      Line 29 – there is lacking two times in a sentence

7.      The authors should correct throughout manuscript the plural of data – it is data again, not datas

8.      The simple summary is almost identical as abstract. The authros should correct this, as the abstract should contain more of the scientific terms

9.      Line 45 – inhibit should be inhibits

10.  Line 99 – there should be a space between 150 and mg

11.  The authors should cite a reference for the ROR formula

12.  What could be a connection between colitis and idelalisib?

Author Response

Reviewer 2 :

Thank you for all your corrections : We made the exchages as follows :

  1. Simple summary – either targeted therapies or targeted therapy
  2. Line 21 there is and without the end of sentence
  3. Line 22 – severe gastro-intestinal is not finished sentence
  4. Line 23 – datas should be changed to data
  5. Lin 24 – give should be changed to gives
  6. Line 29 – there is lacking two times in a sentence
  7. The authors should correct throughout manuscript the plural of data – it is data again, not datas
  8. Line 45 – inhibit should be inhibits
  9. Line 99 – there should be a space between 150 and mg

   The simple summary is almost identical as abstract. The authors should correct this, as the abstract should contain more of the scientific terms

Thank you for this remark. Abstract was changed

The authors should cite a reference for the ROR formula. Reference was added line l51 :

Therapie  2019 April74(2):225-232. doi: 10.1016/j.therap.2019.01.006.

What could be a connection between colitis and idelalisib?

This is an important question. We have described the hypothesis of the pathology of Pi3kinase-induced colitis in our revised manuscript l 370 -371

Round 2

Reviewer 1 Report

Thank you for revising your good manuscript. It has been enhanced for clarity, and it is easier to read. 

The English editing has improved the manuscript, along with the addition and precisions added by the authors.

The whole manuscript has been streamlined, and all concerns have been addressed. 

This reviewer recommends accepting the manuscript in its present form.